# Initiation and Single Dispensing in Cardiovascular and Insulin Medications: Prevalence and Explanatory Factors

**DOI:** 10.3390/ijerph17103358

**Published:** 2020-05-12

**Authors:** Carles Vilaplana-Carnerero, Ignacio Aznar-Lou, María Teresa Peñarrubia-María, Antoni Serrano-Blanco, Rita Fernández-Vergel, Dolors Petitbò-Antúnez, Montserrat Gil-Girbau, Marian March-Pujol, Juan Manuel Mendive, Alba Sánchez-Viñas, Cristina Carbonell-Duacastella, Maria Rubio-Valera

**Affiliations:** 1Research and Development Unit, Institut de Recerca Sant Joan de Déu, 08830 Barcelona, Catalonia, Spain; vilaplanacarles@gmail.com (C.V.-C.); i.aznar@pssjd.org (I.A.-L.); m.gil@pssjd.org (M.G.-G.); albasv10@gmail.com (A.S.-V.); c.carbonell@pssjd.org (C.C.-D.); 2School of Pharmacy, University of Barcelona, 08028 Barcelona, Catalonia, Spain; marianmarch@ub.edu; 3Consortium for Biomedical Research in Epidemiology & Public Health (CIBER en Epidemiología y Salud Pública—CIBERESP), 28029 Madrid, Spain; maitepenarrubia@gmail.com (M.T.P.-M.); aserrano@pssjd.org (A.S.-B.); 4Primary Care Research Institute (IDIAP Jordi Gol), 08007 Barcelona, Spain; rfernandez@ambitcp.catsalut.net (R.F.-V.); juanmmendive@gmail.com (J.M.M.); 5Catalan Institute of Health, 08028 Barcelona, Spain; dolorspetitbo@gmail.com; 6Parc Sanitari Sant Joan de Déu, 08830 Sant Boi de Llobregat, Catalonia, Spain; 7Primary Care Prevention and Health Promotion Research Network, 08007 Barcelona, Catalonia, Spain

**Keywords:** cardiovascular diseases, insulin, adherence, primary care, real-world data, medication initiation

## Abstract

Background: Adherence problems have negative effects on health, but there is little information on the magnitude of non-initiation and single dispensing. Objective: The aim of this study was to estimate the prevalence of non-initiation and single dispensation and identify associated predictive factors for the main treatments prescribed in Primary Care (PC) for cardiovascular disease (CVD) and diabetes. Methods: Cohort study with real-world data. Patients who received a first prescription (2013–2014) for insulins, platelet aggregation inhibitors, angiotensin-converting enzyme inhibitors (ACEI) or statins in Catalan PC were included. The prevalence of non-initiation and single dispensation was calculated. Factors that explained these behaviours were explored. Results: At three months, between 5.7% (ACEI) and 9.1% (antiplatelets) of patients did not initiate their treatment and between 10.6% (statins) and 18.4% (ACEI) filled a single prescription. Body mass index, previous CVD, place of origin and having a substitute prescriber, among others, influenced the risk of non-initiation and single dispensation. Conclusions: The prevalence of non-initiation and single dispensation of CVD medications and insulin prescribed in PC in is high. Patient and health-system factors, such as place of origin and type of prescriber, should be taken into consideration when prescribing new medications for CVD and diabetes.

## 1. Introduction

Cardiovascular disease is highly prevalent and causes high mortality, disability and economic burden [1,2]. After 3.2 years (average follow-up), more than 30% of patients abandon their antidiabetic, antiplatelet, antihypertensive, and lipid-lowering treatments [3]. Non-adherence worsens control of the disease [4] and clinical results in both primary and secondary prevention [5], thereby increasing cardiovascular morbidity and mortality [6] and direct and indirect health costs [7]. Accordingly, controlling cardiovascular medication adherence is important for improving patients’ functional state and the overall sustainability of the system. Most non-adherence studies focus on implementation and discontinuation, yet non-initiation may add to the negative effects of non-adherence in cardiovascular disease and diabetes [8].

The ABC taxonomy defines initiation as the moment “when the patient takes the first dose of a prescribed medication” [9]. Non-adherence thus occurs where there is “late or non-initiation of the prescribed treatment”, a definition that covers patients who do not fill a first prescription, fill it but do not actually take the medication or take it later than expected. This is a comprehensive definition, since it takes into account the patients’ actual behavior vis-à-vis the first prescription. However, collecting patient-reported data in a large population is costly and not free of bias (such as patients’ desirability bias). Taking into account the dispensing process, the International Society for Pharmacoeconomics and Outcomes Research (ISPOR) defines initiation (or initial medication adherence) as “the patient obtaining, for the first time, a new prescription medication” [8]. This definition enables initiation to be ascertained by reference to prescription and dispensing records but has one important limitation: patients can fill the prescription but not take the medication. Hence, non-initiation could be underestimated when using the ISPOR definition. In this study, non-initiation was deemed to occur when a patient did not obtain, for the first time, a new prescription medication.

Single dispensing (or first-fill discontinuation), defined as obtaining only the first unit or container of a new prescription medication, may be capturing patients that fill the prescription but do not take the medication (non-initiation) and patients that try the medication and then discontinue it prematurely (early discontinuation, as defined by the ABC taxonomy) [9]. Although studies usually include patients who filled at least one prescription, few have examined single dispensing of cardiovascular and insulin medications in detail, and even fewer have focused on primary care (PC), the area where these diseases are generally controlled [10,11,12,13].

The literature indicates that 3.3% to 42.3% of PC patients do not initiate medications for prevention and treatment of cardiovascular disease and diabetes [14,15], and up to 28% fill only a single prescription of such medications [11]. In spite of the growing literature on the topic, little is known about the factors that promote and hinder the non-initiation and single dispensing of cardiovascular and insulin medications. A recent systematic review aimed at synthesizing factors that contribute to non-initiation [16] identified nine studies that focused on patients with cardiovascular disease and diabetes. These studies showed that in these patients non-initiation was influenced by factors related to the patient (age and gender), treatment (cost of medication, complexity of treatment and number of concomitant medications) and healthcare provider (lack of patient-doctor communication and distrust of the provider). The studies that focused on single dispensing of cardiovascular or diabetes medications reported that younger, female patients who visited the physician less frequently had a lower probability of filling a second prescription. Qualitative studies indicate that patients’ beliefs and knowledge about the disease and the treatment, emotions, preferences for lifestyle interventions and mistrust of the diagnosis also contribute to non-initiation [17,18].

In a previous study in Catalonia, the prevalence of non-initiation one month after prescription of insulin and treatments for cardiovascular disease was estimated to range from 7.5% through 13.17% [14]. In this study, the prevalence of non-initiation of these treatments 3 months after prescription was between 5.7% and 9.1%, and the prevalence of single dispensing was over 10%. The study also explored the factors that account for non-initiation one month after prescription, and showed that age, country of origin, presence of certain diseases and the characteristics of the prescriber and primary care center influenced the probability of initiation. This analysis included patients who had received a new prescription of a medication contained in the 13 most prescribed and/or costly pharmacotherapeutic subgroups in the public health system, and drugs for acute conditions accounted for over 50% of the prescriptions. Faced with the prescription of a new treatment for a chronic disease, patients may respond with fear or frustration and prefer to delay the start of treatment until they have explored non-pharmacological treatment options [17]. This may delay the start of treatment, which would explain the significant variation observed in the prevalence of non-initiation 1 and 3 months after prescription. Considering that these are chronic treatments, it is relevant to allow for a period of reflection when exploring the prevalence of non-initiation. It is likely that the factors that explain the probability of non-initiation in chronic medications differ from those applicable to acute medications. Furthermore, some factors that might influence non-initiation of treatments for cardiovascular disease and diabetes, such as body mass index (BMI) or existence of previous episodes of cardiovascular disease, were not explored in the previous paper. Since these are chronic treatments, it would be interesting to evaluate the factors underlaying the single dispensing of insulin and treatments for cardiovascular disease.

Better knowledge of both, the prevalence and the explanatory factors of non-initiation and single dispensing of medications for the prevention of cardiovascular disease and diabetes, would help in developing strategies to address these problems. The aim of this study was thus to estimate the prevalence of non-initiation and single dispensing of treatments for cardiovascular disease and diabetes in PC, and to identify the factors accountable for these phenomena.

## 2. Methods

The European Society for Patient Adherence, Compliance and Persistence (ESPACOMP) Medication Adherence Reporting Guidelines (EMERGE) were followed for the reporting of results [19]. The study was approved by the Research Ethics Committees at Jordi Gol Primary Care Research Institute (P14/140) and Sant Joan de Déu Hospital (PIC-111-14).

### 2.1. Setting

The healthcare system in Spain is taxpayer-funded, free at point of delivery and covers both residents and foreign nationals. The system is decentralized, with each of Spain’s 17 Autonomous Regions responsible for local public health, health planning and health-service management [20]. Patients pay a part of the cost of medication in line with the specific medication’s and patient’s characteristics. This co-payment applies to pensioners and non-pensioners alike, and ranges from 0% to 60%, with a 10%-reduced contribution for some medications (most chronic treatments) [21]. Patients in Spain can choose to fill their prescriptions at any pharmacy and are free to switch from one to another on successive visits. Pharmacies submit monthly registries of dispensed medications to the health authorities for reimbursement of the balance due. There are 371 publicly supported PC centers for the 7.5 million inhabitants of Catalonia. The Catalan Health Institute (*Institut Català de la Salut/*ICS) manages most of these centers, with other healthcare providers managing the remainder. The 294 PC centers managed by the Catalan Health Institute cover 80% of the region’s population (5.8 million) [22].

PC is the first point of contact with the health system. Each person is allocated a general practitioner (GP) who generates all prescriptions, except when on leave or when a substitute or resident GP conducts the consultation.

### 2.2. Design

This was a secondary analysis of a previous study which focused on the population with cardiovascular diseases and diabetes [14]. A cohort study based on electronic records of routine clinical practice was conducted (Real-World Data). The data source used was the Information for the Development of Research in Primary Care Database (*Sistema d’Informació per al Desenvolupament de la Investigació en Atenció Primària/*SIDIAP database) [22], which contains computerized PC clinical history information from the Catalan Health Institute (80% of the Catalan population) and information on pharmaceutical billing. The SIDIAP database is managed by public healthcare authorities, is anonymous, encoded and secure and meets all current legal requirements. All the information used for analysis purposes was recorded between 2011 and 2014, and it was gathered from the database in 2015. This information contains patient, GP and PC center information, along with information on prescription and whether or not a given prescription has been dispensed. Comparison of the validity of SIDIAP data for the study of cardiovascular diseases to that of data from the REGICOR2000 study [23] showed a high level of validity of the data and good representativeness of the population in the SIDIAP database for use in epidemiological studies of cardiovascular disease [23].

This was a secondary analysis of a study on non-initiation in PC [14]. Patients that were newly prescribed a medication included in the list of the 10 most prescribed pharmacotherapeutic subgroups (based on the Anatomical Therapeutic Chemical (ATC) classification system) or the seven most costly pharmacotherapeutic subgroups in 2014 for the Catalan Healthcare Institute were included in the original study [14]. “Insulins and analogues for injection, long-acting” (ATC A10AE), “Platelet aggregation inhibitors excluding heparin” (ATC B01AC), “Angiotensin-converting enzyme inhibitors (ACEIs), plain” (ATC C09AA) and “HMG CoA reductase inhibitors” (ATC C10AA) were among the most prescribed and costly pharmacotherapeutic subgroups and were selected for the study of initiation and single dispensing in cardiovascular disease and diabetes.

Index prescriptions were defined as new prescriptions of one medication from the four pharmacotherapeutic subgroups under study. The study included all PC patients aged ≥15 years with prescriptions across the period June 2013–July 2014. In the Catalan health system, prescriptions have a start and end date which covers the entire treatment period, and medications are automatically made available to patients as they need them. In the case of medications for chronic conditions, prescriptions are usually activated for a 12-month period and renewed every year. Following the recommendations of GPs, prescriptions were considered new if there was no other active prescription from the same pharmacological subgroup recorded in the three months prior to the index prescription. There were no further inclusion criteria.

### 2.3. Variables

All the information used for the analysis was gathered from the database, which contains information on prescription and dispensing, as well as sociodemographic and biomedical data relating to the patients, GPs and PC centers. Data were collected for all patients over a period of 12 months preceding the index prescription. Prescriptions were followed-up 6 months after the index prescription. Prescriptions not filled at the pharmacy 3 and 6 months after prescription (sensitivity analysis) were deemed to be non-initiated (in the primary study non-initiation 1 and 3 months after prescription was reported). If a single container was picked up during the first three months (in the primary study, the first month was considered) and no other container was collected in a period of up to six months after the first prescription, this was regarded as single dispensing.

As in the primary study, the following factors were assessed as patient explanatory factors: age, gender, place of origin (continent of birth), socioeconomic status and pathologies at the date of prescription. In this study, the following factors were also assessed: previous cardiovascular disease (recent and established), body mass index (BMI), existence of previous new general prescriptions (of any of the 10 most prescribed or the seven most costly pharmacotherapeutic subgroups in 2014), existence of previous new cardiovascular prescriptions and medical and nursing visits in PC.

Information on socioeconomic status was based on the MEDEA index [24], which was estimated using the weighted sum of five census-based socioeconomic indicators (unemployment rate, manual workers, temporary workers, illiterate adults (or less than basic, compulsory education) and school drop-outs from among the population aged <16 years). Through a link to the census data, SIDIAP allocates patients to one of the five urban levels of the MEDEA index (with 1 representing the lowest and 5 the highest socioeconomic status). There is no information on the socioeconomic status of patients living in rural areas since the MEDEA was only validated for urban populations.

To ensure anonymity, only the most prevalent diseases related to the treatments under study were obtained. Pathologies were grouped into the following categories: allergy, pain (including arthritis, arthrosis, rheumatological diseases and back pain), respiratory (including asthma and chronic obstructive pulmonary disease), physical disability (including blindness, urinary incontinency and hypoacusia), cardiovascular conditions (arterial hypertension, dyslipidemia, use of substances like alcohol or tobacco (alcohol or tobacco use was based on the qualitative appreciation of GPs; alcohol or tobacco use, though not cardiovascular conditions, are included as cardiovascular risk factors) and cardiovascular diseases), mental (depression; schizophrenia and neurotic, stress and somatic symptom disorders), neurological (neuropathy, epilepsy and migraine), diabetes mellitus (types 1 and 2), digestive (cirrhosis, chronic constipation, hiatus hernia, peptic ulcer, dyspepsia and gastroesophageal reflux disease) and thyroid-related diseases.

Previous cardiovascular disease (myocardial infarction, cardiac insufficiency, peripheral arterial disease, stroke or ischemic cardiopathology) was classified as follows: no previous cardiovascular disease, recent cardiovascular disease (≤6 months before the prescription) and established cardiovascular disease (>6 months before the prescription).

The existence of previous new general prescriptions took into account any new prescriptions issued in the 12 months preceding the index prescription for proton pump inhibitors, insulin, antiplatelets, ACEIs, HMG CoA reductase inhibitors, penicillin, propionic acid derivatives, anilides, antiepileptics, benzodiazepine, selective serotonin reuptake inhibitors, adrenergics in combination or anticholinergics. Similarly, the existence of previous new cardiovascular prescriptions took into account new prescriptions of the treatments under study issued in the 12 months preceding the index prescription.

With regard to the prescribing GP, the variables of age, gender and type of prescriber (allocated or substitute GP) were assessed. Health centers were classified according to whether or not they were teaching hospitals.

### 2.4. Analysis

The basic unit of analysis was the prescription. The prevalence of non-initiation and single dispensing was estimated for each pharmacological subgroup.

To manage missing data (BMI (30.8%), prescriber sex (9.7%), socioeconomic status (4.2%) and place of origin (41%)), simple imputation by chained equations was used with logistic regression and ordinal logistic models using all the available data (2011–2014) [14]. This imputation method has been previously used with satisfactory results [14,21,25,26]. For BMI, a truncated regression model with a lower limit of 10 was used.

In order to identify the explanatory factors of initiation and single dispensing, two multivariate multilevel logistic regressions were conducted (using initiation 3 months after prescription and single dispensing as dependent variables), overall and for each pharmacotherapeutic group. By way of a sensitivity analysis, the explanatory factors of initiation 6 months after prescription were assessed.

As recommended by Mickey and Greenland, the selection of variables for inclusion in the multivariate analysis was based on both the change-in estimate criterion and significance testing methods [27]. Results of the bivariate analysis are shown as Appendix A.

First, bivariate multilevel logistic regression models were applied to select the variables to be included in the multivariate model using statistical significance and effect size criteria. Variables that showed a statistically significant association (*p*-value < 0.05) (significance testing) [27] and had an odds ratio (OR) larger than set cut-off points (OR >1.1 or <0.9, in categorical variables; and OR > 1.11 or <0.99 in continuous variables) (change-in estimate criterion) [27] were included in the multivariate model. The results of the bivariate analysis are shown in Appendix A.

For the selection of interactions to be included in the multivariate model, a more conservative approach was adopted to avoid over-adjusting the model. Interactions considered clinically relevant were tested and included in the multivariate logistic regression model in any case where they reached a statistically significant association (*p*-value < 0.001) and the estimate was large enough to reverse the value of the OR when the interaction was present. None of the interactions tested fulfilled the criteria for inclusion in the final multivariate model.

As insulins and statins have a single indication (diabetes and dyslipidemia, respectively), the presence/absence of these diseases was excluded from their respective models.

All analyses were performed using Stata MP13.0 (StataCorp LLC, Lakeway Drive College Station, TX, USA).

### 2.5. Availability of Data

The data that support the findings of this study are available from the Information for the Development of Research in Primary Care Database (*Sistema d’Informació per al Desenvolupament de la Investigació en Atenció Primària/*SIDIAP database), but restrictions apply to the availability of these data, since it contains patient information.

## 3. Results

Table 1 shows the sociodemographic characteristics of the sample, which included 8270 prescriptions for insulins, 34,139 for antiplatelets, 74,346 for ACEIs, and 69,602 for statins, issued to 169,143 patients at 287 PC centers.

### 3.1. Prevalence of Non-initiation and Single Dispensing

Table 2 shows the ratios for non-initiation and single dispensing for the four pharmacological subgroups. At three months, the non-initiation ratio ranged from 5.7% (ACEIs) to 9.1% (antiplatelets), while the single dispensing ratio ranged from 10.6% (statins) to 18.4% (ACEIs).

### 3.2. Non-initiation Explanatory Factors

Figure 1 summarizes the explanatory factors of non-initiation at 3 and 6 months (sensitivity analysis) after the date of prescription and single dispensing. Table 3 and Appendix A show the results of the multivariate multilevel logistic regression models for factors explaining non-initiation at 3 and 6 months after prescription, respectively, for each pharmacological subgroup.

The factors that accounted for non-initiation at 6 months after prescription were the same as those that accounted for non-initiation at 3 months after prescription with a few exceptions (Figure 1).

A higher BMI and presence of hypertension decreased the probability of non-initiation of insulins, antiplatelets, ACEIs, and statins. In specific pharmacological subgroups, the following factors lowered the risk of non-initiation: older age; higher socioeconomic status; having at least one new general prescription in the last 12 months; higher number of visits to the GP or nurse; pain; physical disability; dyslipidemia; mental disorders; diabetes; digestive disorders; and an established cardiovascular disease.

A foreign place of origin and an established cardiovascular disease increased the risk of non-initiation in all four pharmacological subgroups.

A recent cardiovascular disease and receiving the prescription from a substitute or resident GP increased the risk of non-initiation of some medications and decreased the risk of others.

### 3.3. Single Dispensing Explanatory Factors

Figure 1 and Table 4 show the factors that affect the probability of filling a single prescription for each pharmacological subgroup.

The explanatory factors of single dispensing coincided with those of non-initiation, with the following exceptions: women had a higher risk of single dispensing of antiplatelets and ACEIs; recent and/or established cardiovascular diseases decreased the risk of single dispensing of antiplatelets, ACEIs and statins; having at least one new general prescription in the last 12 months and receiving the prescription in a non-teaching PC center increased the risk of single dispensing; and lastly, receiving the prescription from a substitute or resident GP increased the risk of single dispensing of antiplatelets, ACEIs and statins.

## 4. Discussion

Non-initiation and single dispensing of treatments prescribed in PC in Catalonia for cardiovascular disease and diabetes are relatively common. Overall, non-initiation of these treatments was slightly higher than in other European countries [15,28] and lower than in Canada [29]. The magnitude of single dispensing of these treatments in Catalonia was in line with published data on similar populations [10,11,12], with the exception of some studies undertaken in the USA, where single dispensing rates of up to 23% and 26% can be observed for insulins [30] and antihypertensive drugs [31], respectively. Differences between settings could be due to the organization of the health system and clinical costs (which include the cost of diagnosis, prescription and medication assumed by the patient). They could also be explained by differences in the characteristics of the study population, which might not have been comparable in terms of severity and prognosis of the episode. Another possible explanation lies in methodological differences between studies in terms of pharmacological subgroups examined, previous period without medication, duration of follow-up and sources of information [8].

As expected, the factors that account for non-initiation of insulins and treatments for cardiovascular disease differ from those identified in the primary study [14], which explored factors that explained non-initiation of treatments for acute and chronic physical and mental disorders. In line with previous studies focusing on treatments for cardiovascular disease and/or diabetes, younger age, lower socioeconomic status, place of origin, number of concomitant prescriptions, and distrust in the provider (prescription held by a substitute/resident physician) increased the risk of non-initiation [17,18,28,32], while younger age and female gender increased the risk of single dispensing [10,30]. In contrast, this study also identified explanatory factors of non-initiation and single dispensing not described in the literature, such as BMI and concomitant diseases. Moreover, to the best of our knowledge, this is the first study to assess the explanatory factors of both non-initiation and single dispensing. The sensitivity analysis showed that explanatory factors of non-initiation 6 months after prescription are consistent with those of non-initiation 3 months after prescription. However, the factors associated with non-initiation and the direction of the association vary within medication groups. Hence, the results cannot be generalized to all types of medications for cardiovascular disease and diabetes.

Greater BMI was associated with a lower probability of non-initiation in all groups of medications and for single dispensing of insulins and statins. Patients with a diagnosis of hypertension, dyslipidemia, diabetes, pain, mental disorders or physical disabilities also had a lower risk of non-initiation and/or single dispensing. Although there is some evidence to show that a higher chronic disease score and previous cardiovascular diseases increase the likelihood of filling more than a single prescription [33], the only published study to assess BMI did not find that it affected initiation [13]. Patients with obesity and overweight have a keener perception of the need for pharmacotherapy [34] and disease awareness, and perception of severity has been shown to improve adherence [17,35]. This would explain why greater BMI and certain diagnoses (pain, physical disability, hypertension, dyslipidemia, diabetes and recent cardiovascular disease) increase the likelihood of initiating and/or filling more than one prescription for cardiovascular disease and diabetes.

Along these same lines, the fact of having a recent or established cardiovascular disease (including myocardial infarction, cardiac insufficiency, peripheral arterial disease, stroke or ischemic cardiopathology) decreased the risk of single dispensing of antiplatelets, ACEIs and statins. However, when the relationship between a recent or established cardiovascular disease and non-initiation was explored in depth, contradictory results were observed. Having an established cardiovascular disease increased the risk of non-initiation of antiplatelets, ACEIs and statins. This contradicts adherence models which suggest that perception of greater severity increases the likelihood of adequate adherence [17,35], and also stands in contrast to the results observed for BMI, hypertension, dyslipidemia and diabetes. A possible explanation is that months after the event the patient may attach less importance to the diagnosis. Hussain et al. found that adherence in post myocardial infarction patients declined significantly over time [36]. This could also be due to polymedication and negative past experience with the treatment [17]. Another contradictory result is that, whereas having a recent cardiovascular disease increased the risk of non-initiation of ACEIs, it decreased the risk of non-initiation of statins. Previous studies have reported lower levels of adherence to ACEIs than to statins after a cardiovascular event [37]. The patient may underestimate the danger of hypertension and fail to link it to cardiovascular disease [38]. Another possible explanation is that the patient is aware that ACEIs are generally not recommended after a cardiovascular event [39]. To confirm this finding, future studies should explore the influence of a recently established cardiovascular disease, and qualitative studies with patients should be conducted in order to better understand the influence of cardiovascular disease on initiation.

Urticaria was an insulin non-initiation risk factor with a high effect size. This could be due to fear of pain and potential changes to the skin [32]. Health professionals should bear this in mind when prescribing insulins. Although they are related to poorer self-care [40], and no prior study has found such an association, mental disorders nonetheless increased the probability of initiating insulins and antiplatelets and the likelihood of filling more than a single prescription for antiplatelets. Major depression has been linked to frequent attendance in Spanish PC [41]. The higher probability of people with mental disorders initiating a treatment for cardiovascular disease and insulins can be related to the higher number of contacts with their GPs. This finding calls for further research in the future.

As previously described, lower socioeconomic status and foreign place of origin were risk factors for non-initiation [15]. The cost of treatment is a key factor in adherence, which would explain the influence of socioeconomic status. With respect to place of origin, coming from the Americas stands out as a non-initiation and single dispensing risk factor. The language barrier should be a minor problem, as in the study context the group in question is mainly composed of people of Central and South American origin. This could be due to differences in health systems in the countries of origin and to cultural issues [42]. The South American population in Catalonia usually demands a large number of medical procedures [43], which could indicate a lack of trust in the system or a lack of understanding between patients and health professionals for cultural reasons.

In line with previous studies [13] and with models that identify the GP-patient and nurse-patient relationship as vital factors in adherence [17,35], the number of previous medical and nursing visits and the type of prescriber influence the probability of initiating. A greater number of visits can lead to a strong patient-health professional-alliance, and this may in turn affect adherence. Similarly, receiving a prescription from a substitute GP reduces confidence in the prescription [44]. This is not the case with antiplatelets, for which the likelihood of initiating is greater when a resident or substitute GP generates the prescription. It may be that these prescriptions are issued as a matter of urgency or due to great severity, which would influence the tendency to initiate the medication prescribed, though the data do not allow for identification of all cases in which the prescription was made out by the assigned GP. Results from previous studies indicate that the pharmacist is an important source of information for the patient and plays an important role in the decision to initiate a treatment [17]. In this study, the influence of the dispensing pharmacist could not be assessed. In the future, the role played by these factors in non-initiation should be examined more closely.

The factors that explain single dispensing are similar to, but not the same as, those that explain non-initiation. This supports the hypothesis that single dispensing represents two different behaviors as defined by the ABC taxonomy, i.e., non-initiation and early discontinuation. While previous studies estimated both non-initiation and single dispensing of antihypertensive, lipid lowering and antidiabetic medications [12,31], they did not, however, explore the factors contributing to such non-initiation and single dispensing.

Given that the protective effect of medications for the prevention of cardiovascular disease occurs over the long term, it would be of value to conduct a medium- and long-term follow-up of non-initiating patients in order to better understand the clinical and economic consequences of non-initiation and single dispensing of these treatments. To date, with few exceptions, pharmacological adherence has been studied in a piecemeal way, focusing on the persistence and degree of adherence to initiated prescriptions [12,31]. Studying initiation, discontinuation and implementation jointly would yield a more realistic picture of the situation.

Three interventions have been implemented to address the problem of non-initiation of cardiovascular treatments [45,46,47], though they only deal with the question of forgetting to fill the prescription, thereby overlooking the fact that this is a multifactorial issue which strongly depends on patients’ perceptions. More intense, multicomponent interventions, such as a motivational interview or a face-to-face consultation with a health professional, could be more effective [47]. In addition, the intervention should also consider the complexity and structure of health systems [47] and adopt a multidisciplinary approach to this type of patient [47].

### 4.1. Strengths and Limitations

The main strength of this study is its representativeness. Even so, non-initiation may have been underestimated, as some patients may have collected the medication but not initiated it. These patients should, in theory, be included in the single dispensing group, though it was impossible for us to determine which patients took the medication and which did not in cases where a single prescription was filled.

Although the data was analyzed retrospectively, all the factors explored (sociodemographic and clinical characteristics of patients and characteristics of the GPs and PC centers) preceded (non-)initiation. This means that both the temporality and direction of causality are clear, and there can be no doubt that what we are describing are indeed explanatory factors.

It would have been of interest to assess initiation and single dispensing in oral antidiabetic drugs. However, these were not among the most prescribed and costly pharmacotherapeutic subgroups, and information on these medications was not available.

As its aim is the registration of clinical practice, the quality of the Real-World Data register is a standard limitation. Only 45% of patients who received a new prescription for a statin had an active diagnosis of dyslipidemia. This may be explained in part by the inappropriateness of the prescription but is more likely to indicate a registry failure. Furthermore, socioeconomic status reflects the reality of the residential area rather than that of the patient [22].

Another limitation of using information from Real-World Data is that there could be confounding factors which have not been accounted for. Important determinants of adherence were not measured. The results of qualitative studies suggest that initial medication adherence is influenced by the patients’ beliefs, preferences, knowledge and emotional reactions [17,18,35]. The cost of treatment and the level of co-payment are also important factors [21]. Similarly, the patients’ context and interaction with their GPs and other healthcare providers, such as pharmacists, could also influence adherence. The influence of these factors was not explored.

The BMI of our study population was higher than that of the general population in Spain. These differences may be due to the fact that our study included patients suffering from diabetes and cardiovascular disease. Sample populations with cardiovascular disease and diabetes in previous studies in Catalonia presented similar BMI values [48].

The results of the present paper are based on data that is five years old. To the best of our knowledge, however, this study not only reflects the most up-to-date data available on initiation and single dispensing in Spain but is one of the most recent studies overall. Explanatory factors of non-initiation and single dispensing are not expected to have changed considerably since the data were obtained.

### 4.2. Practical Implications

This study identifies a series of factors known by GPs to increase the risk of low adherence in initiating treatment for cardiovascular disease and diabetes. These factors should be borne in mind as warning signs when new prescriptions are issued for these medications in PC.

It would be advisable to draw up strategies aimed at improving adherence to initiation of treatment with cardiovascular and diabetes medications and to monitor dispensing closely so as to ensure that patients initiate new prescriptions.

As in previous studies [13,14,18,42], deficiencies in care of the most vulnerable populations were identified, with lower rates of initiation and single dispensing among patients from other countries and poorer initiation rates among those with the lowest socioeconomic status. Ensuring equity in health care is a pillar of public health in Spain; specific, intensive strategies for these populations should thus be developed.

The role of PC nursing is fundamental in the management and control of diabetes and initiation of treatment with insulins. The results of this study support strategies that include nursing in the management of cardiovascular disease [49]. The role of community pharmacists should also be assessed, as they already play an important role in cardiovascular treatment dispensing and adherence, especially when it comes to initial prescriptions.

It would also be advisable to raise public awareness about these medications and their relationship with cardiovascular disease. Likewise, improving the approach to hypertension should be a priority in PC. Strategies that increase patient trust in substitute GPs would also be useful.

### 4.3. Future Research

Future studies should examine adherence to medication for cardiovascular disease as a whole, taking into account initiation, implementation and discontinuation. The influence of factors pertaining to the GP-patient and nurse-patient relationship should be studied in greater detail, assessing the role of substitute and resident GPs in non-initiation and single dispensing.

Few studies have explored single dispensing when exploring non-initiation. The prevalence of single dispensing in cardiovascular disease and insulin is high, and further studies are required to examine this in detail. It would be of interest to ascertain the extent to which single dispensing can be explained by non-initiation and early discontinuation.

Studies examining the effectiveness and efficiency of interventions directed at increasing initiation are needed, in order to guide future interventions targeting patients in PC and in other contexts.

## 5. Conclusions

The prevalence of non-initiation and single dispensing of cardiovascular and insulin medication treatments is high in Catalonia. The following factors increase the likelihood of initiating treatment: older age; higher BMI; higher socioeconomic status and diagnosis of pain, hypertension, dyslipidemia, diabetes, recent cardiovascular disease and mental disorders. Among factors that reduce the probability of initiating treatment, the following stand out: foreign place of origin, established cardiovascular disease, urticaria and substitute or resident prescriber. With some exceptions, the explanatory factors of single dispensing are similar to those of non-initiation, though more studies are needed for a proper understanding of single dispensing.

## Figures and Tables

**Figure 1 ijerph-17-03358-f001:**
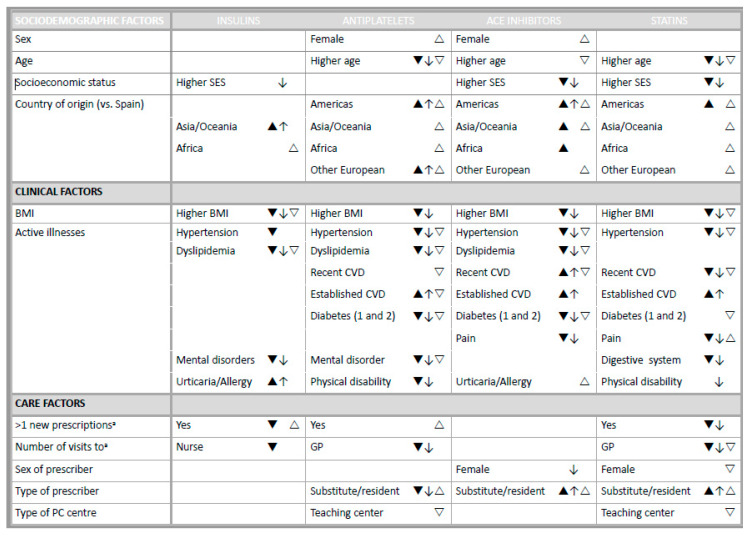
Factors that increase/decrease the probability of non-initiation after 3 months (▲/▼) and 6 months (↑/↓) and of single dispensing within a 3-month period (△/▽).

**Table 1 ijerph-17-03358-t001:** Sociodemographic characteristics of the sample.

**PATIENTS; *n* = 169,143**	**Insulin** ***n* = 8223**	**Antiplatelet** ***n* = 33,921**	**ACEI** ***n* = 73,741**	**Statin** ***n* = 69,043**
Sex, Female, *n* (%)	3848 (46.80)	17,087 (50.37)	36,583 (49.61)	36,334 (52.63)
Age, mean ± SD	67.50 ± 15.04	70.47 ± 13.94	65.65 ± 14.28	62.84 ± 12.72
Socioeconomic status, *n* (%)				
Urban 1 (lowest SES)	1160 (14.11)	6101 (17.99)	11,717 (15.89)	11,945 (17.30)
Urban 2	1271 (15.46)	5552 (16.37)	12,019 (16.30)	11,415 (16.53)
Urban 3	1352 (16.44)	5459 (16.09)	12,360 (16.76)	11,378 (16.48)
Urban 4	1395 (16.96)	5491 (16.19)	12,231 (16.59)	11,041 (15.99)
Urban 5 (highest SES)	1509 (18.35)	4982 (14.69)	11,470 (15.55)	10,040 (14.54)
Rural	1536 (18.68)	6336 (18.68)	13,944 (18.91)	13,224 (19.15)
Place of origin, *n* (%)				
Spain	7446 (90.55)	31,555 (93.02)	66,075 (89.60)	62,315 (90.26)
Americas	281 (3.42)	995 (2.93)	3022 (4.10)	2915 (4.22)
Asia/Oceania	114 (1.39)	269 (0.79)	963 (1.31)	758 (1.10)
European outside Spain	117 (1.42)	561 (1.65)	1621 (2.20)	1555 (2.25)
Africa	265 (3.22)	541 (1.59)	2060 (2.79)	1500 (2.17)
Body mass index, mean ± SD	29.53 ± 5.56	28.68 ± 5.09	29.31 ± 5.11	28.70 ± 4.87
New prescriptions ^a^, *n* (%)				
≥1 medications	1487 (18.08)	6251 (18.43)	14,225 (19.29)	13,218 (19.14)
≥1 cardiovascular/diabetes medications	289 (3.51)	1030 (3.04)	1606 (2.18)	2095 (3.03)
Number of visits ^a^, mean ± SD				
Visits to GP	8.34 ± 6.68	7.05 ± 5.92	6.52 ± 5.54	5.96 ± 5.01
Visits to nurse	7.89 ± 9.31	4.62 ± 7.73	4.43 ± 6.84	3.43 ± 6.38
**PRESCRIPTIONS; *n*= 186,357**	**Insulin** ***n* = 8270**	**Antiplatelet** ***n* = 34,139**	**ACEI** ***n* = 74,346**	**Statin** ***n* = 69,602**
**Illnesses at the moment of prescription, *n* (%)**				
Pain	3658 (44.23)	16,919 (49.56)	34,696 (46.67)	31,389 (45.10)
Respiratory	1145 (13.85)	4530 (13.27)	8174 (10.99)	6331 (9.10)
Physical disability ^b^	3015 (36.46)	12,261 (35.91)	21,526 (28.95)	16,410 (23.58)
Cardiovascular				
Hypertension	5477 (66.23)	20,996 (61.50)	55,916 (75.21)	32,799 (47.12)
Dyslipidemia	3206 (38.77)	10,922 (31.99)	21,998 (29.59)	31,451 (45.19)
Recent CVD (≤6 months)	196 (2.37)	2697 (7.90)	1949 (2.62)	1948 (2.80)
Established CVD (>6 months)	2098 (25.37)	6372 (18.66)	9333 (12.55)	6685 (9.60)
Alcohol and tobacco use ^c^	2272 (27.47)	8456 (24.77)	18,912 (25.44)	19,613 (28.18)
Neurological	1057 (12.78)	3529 (10.34)	7317 (9.84)	7024 (10.09)
Mental disorders	2192 (26.51)	9456 (27.70)	19,589 (26.35)	19,724 (28.34)
Diabetes (1 and 2)	7698 (93.08)	8967 (26.27)	15,220 (20.47)	14,755 (21.20)
Digestive system disorder	1671 (20.21)	7638 (22.37)	14,329 (19.27)	12,498 (17.96)
Urticaria/allergy	145 (1.75)	786 (2.30)	1705 (2.29)	1652 (2.37)
Hyper/hypothyroidism	745 (9.01)	2970 (8.70)	6034 (8.12)	6392 (9.18)
Number of comorbidities ^d^, mean ± SD	3.43 ± 1.47	2.79 ± 1.48	2.61 ± 1.43	2.47 ± 1.42
**PESCRIBER; *n* = 4995**	**Insulin** ***n* = 3125**	**Antiplatelet** ***n* = 4272**	**ACEI** ***n* = 4612**	**Statin** ***n* = 4541**
Sex, female, *n* (%)	2025 (64.80)	2867 (67.11)	3118 (67.61)	3043 (67.01)
Age, mean ± SD	48.17 ± 9.98	47.06 ± 10.53	46.51 ± 10.68	46.64 ± 10.68
Type of prescriber, *n* (%)				
Assigned GP	2902 (92.86)	3716 (86.99)	3884 (84.22)	3858 (84.96)
Substitute/resident GP	223 (7.14)	556 (13.01)	728 (15.78)	683 (15.04)
**CENTER; *n* = 287**	**Insulin** ***n* = 283**	**Antiplatelet** ***n* = 285**	**ACEI** ***n* = 285**	**Statin** ***n* = 286**
Teaching centers, *n* (%)	72 (25.44)	72 (25.26)	72 (25.26)	72 (25.17)

ACEI: Angiotensin-converting enzyme inhibitors; CVD: Cardiovascular disease; GP: General practitioner; SES: Socioeconomic status. ^a^ In the 12 months preceding the index prescription. ^b^ Physical disabilities include blindness, urinary incontinency and hypoacusia. ^c^ Alcohol or tobacco use was based on the qualitative appreciation of GPs; alcohol or tobacco use, though not cardiovascular conditions, are included as cardiovascular risk factors. ^d^ Number of comorbidities includes all the active illnesses listed in the table.

**Table 2 ijerph-17-03358-t002:** Non-initiation and single dispensing rates, *n* (%).

Pharmacological Subgroup (ATC Code)	Non-Initiation after 3 Months	Non-Initiation after 6 Months	Single Dispensing
Insulin (A10AE)	643 (7.78)	507 (6.13)	1207 (14.59)
Antiplatelet (B01AC)	3111 (9.11)	2685 (7.86)	4639 (13.59)
ACEI (C09AA)	4210 (5.66)	3678 (4.95)	13,691 (18.42)
Statin (C10AA)	4693 (6.74)	4056 (5.83)	7362 (10.58)

ACEI: Angiotensin-converting enzyme inhibitors.

**Table 3 ijerph-17-03358-t003:** Explanatory factors of non-initiation after 3 months based on multivariate multilevel models.

	Insulin^a^ *n* = 8223		Antiplatelet^b^ *n* = 33,921		ACEI^c^ *n* = 73,741		Statin^d^ *n* = 69,043	
	OR 95% CI	*p*-Value	OR 95% CI	*p*-Value	OR 95% CI	*p*-Value	OR 95% CI	*p*-Value
**Constant**	0.458 (0.25;0.84)	0.012	0.531 (0.39;0.72)	0.001	0.145 (0.12;0.18)	0.001	0.326 (0.25;0.42)	0.001
**Female patient (vs. male)**	0.994 (0.84;1.18)	0.950	—	—	—	—	0.994 (0.93;1.06)	0.845
**Patient’s age (cont.)**	0.998 (0.99;1.00)	0.492	**0.987** **(0.98;0.99)**	**0.001**	—	—	**0.988** **(0.98;0.99)**	**0.001**
**Patient’s SES^a^**								
**Urban 1 (lowest SES)**	Ref.		Ref.		Ref.		Ref.	
**Urban 2**	0.842 (0.63;1.13)	0.255	1.024 (0.89;1.17)	0.730	**0.812** **(0.73;0.91)**	**0.001**	**0.807** **(0.73;0.89)**	**0.001**
**Urban 3**	0.741 (0.55;1.00)	0.052	0.982 (0.85;1.13)	0.803	**0.762** **(0.68;0.85)**	**0.001**	**0.768** **(0.69;0.85)**	**0.001**
**Urban 4**	0.763 (0.56;1.03)	0.079	0.898 (0.78;1.04)	0.150	**0.683** **(0.61;0.77)**	**0.001**	**0.703** **(0.63;0.78)**	**0.001**
**Urban 5 (highest SES)**	0.830 (0.62;1.12)	0.216	0.931 (0.80;1.08)	0.361	**0.698** **(0.62;0.79)**	**0.001**	**0.696** **(0.62;0.78)**	**0.001**
**Rural**	0.887 (0.66;1.19)	0.422	1.151 (1.00;1.32)	0.049	**0.798** **(0.71;0.89)**	**0.001**	**0.798** **(0.72;0.89)**	**0.001**
**Patient’s place of origin**								
**Spain**	Ref.		Ref.		Ref.		Ref.	
**Americas**	1.148 (0.76;1.74)	0.516	**1.335** **(1.10;1.63)**	**0.004**	**1.265** **(1.09;1.47)**	**0.002**	**1.225** **(1.07;1.40)**	**0.003**
**Asia/Oceania**	**1.864** **(1.08;3.21)**	**0.024**	1.173 (0.81;1.70)	0.398	**1.354** **(1.05;1.74)**	**0.019**	1.180 (0.91;1.53)	0.209
**Europe outside Spain**	1.238 (0.68;2.26)	0.488	**1.632** **(1.30;2.06)**	**0.001**	1.192 (0.97;1.46)	0.087	1.134 (0.94;1.36)	0.178
**Africa**	1.267 (0.84;1.91)	0.261	1.143 (0.87;1.51)	0.345	**1.234** **(1.04;1.48)**	**0.025**	0.910 (0.74;1.12)	0.375
**BMI (cont.)**	**0.971** **(0.95;0.99)**	**0.001**	**0.979** **(0.97;0.99)**	**0.001**	**0.981** **(0.98;0.99)**	**0.001**	**0.991** **(0.99;1.00)**	**0.009**
**≥1** **New general prescriptions ^a^** **(vs. none)**	**0.756** **(0.60;0.96)**	**0.022**	0.919 (0.83;1.02)	0.119	—	—	**0.848** **(0.78;0.93)**	**0.001**
**≥1 CV/diabetes new prescriptions^a^ (vs. none)**	—	—	—	—	—	—	1.028 (0.83;1.27)	0.794
**Number of visits^a^ (cont.)**								
**Visits to GP**	0.995 (0.98;1.01)	0.580	**0.978** **(0.97;0.99)**	**0.001**	—	—	**0.982** **(0.97;0.99)**	**0.001**
**Visits to nurse**	**0.985** **(0.97;1.00)**	**0.030**	0.993 (0.99;1.00)	0.052	—	—	0.998 (0.99;1.00)	0.543
**Active illnesses**								
**Pain**	0.909 (0.76;1.09)	0.305	1.000 (0.92;1.08)	0.993	**0.850** **(0.80;0.91)**	**0.001**	**0.919** **(0.86;0.98)**	**0.011**
**Respiratory**	—	—	0.950 (0.84;1.07)	0.401	—	—	0.963 (0.86;1.08)	0.510
**Physical disability ^b^**	—	—	**0.873** **(0.80;0.96)**	**0.003**	—	—	0.925 (0.85;1.00)	0.058
**Cardiovascular conditions**								
**Hypertension**	**0.751** **(0.62;0.91)**	**0.004**	**0.813** **(0.75;0.88)**	**0.001**	**0.800** **(0.75;0.86)**	**0.001**	**0.767** **(0.72;0.82)**	**0.001**
**Dyslipidemia**	**0.723** **(0.60;0.87)**	**0.001**	**0.856** **(0.78;0.93)**	**0.001**	**0.859** **(0.80;0.92)**	**0.001**	—	—
**Recent CVD (≤ 6 months)**	—	—	0.909 (0.77;1.07)	0.256	**1.268** **(1.05;1.53)**	**0.014**	**0.625** **(0.49;0.79)**	**0.001**
**Established CVD (>6 months)**	—	—	**3.155** **(2.90;3.44)**	**0.001**	**2.265** **(2.09;2.45)**	**0.001**	**1.504** **(1.36;1.66)**	**0.001**
**Mental disorders**	**0.801** **(0.65;0.99)**	**0.041**	**0.839** **(0.77;0.92)**	**0.001**	0.969 (0.90;1.04)	0.400	—	—
**Neurological**	—	—	—	—	—	—	—	—
**Diabetes (1 and 2)**	—	—	**0.817** **(0.74;0.90)**	**0.001**	**0.871** **(0.80;0.95)**	**0.001**	1.000 (0.92;1.08)	0.994
**Digestive system disorder**	0.812 (0.64;1.02)	0.078	0.980 (0.89;1.08)	0.684	—	—	**0.917** **(0.84;1.00)**	**0.046**
**Urticaria/allergy**	**2.110** **(1.27;3.5)**	**0.004**	—	—	—	—	—	—
**Hyper/hypothyroidism**	—	—	0.939 (0.81;1.08)	0.389	—	—	0.927 (0.83;1.04)	0.182
**Substitute/resident GP (vs. other)**	—	—	**0.805 (0.66;0.99)**	**0.036**	**1.212 (1.05;1.40)**	**0.009**	**1.204 (1.05;1.38)**	**0.006**

ACEI: Angiotensin-converting enzyme inhibitors; cont.: continuous variable; CVD: Cardiovascular disease; GP: General practitioner; SES: Socioeconomic status. **Bold** numbers indicate statistically significant results. “—” indicates that the factor was not selected to be included in the multivariate analysis based on results from the bivariate analysis. ^a^ In the 12 months preceding the index prescription. ^b^ Physical disabilities include blindness, urinary incontinency and hypoacusia.

**Table 4 ijerph-17-03358-t004:** Explanatory factors of single dispensing within a 3-months period based on multivariate multilevel models.

	Insulin ^a^ *n* = 8223		Antiplatelet ^b^ *n* = 33,921		ACEI ^c^ *n* = 73,741		Statin ^d^ *n* = 69,043	
	OR 95% CI	*p*-Value	OR 95% CI	*p*-Value	OR 95% CI	*p*-Value	OR 95% CI	*p*-Value
**Constant**	0.378 (0.26;0.53)	0.001	1.048 (0.82;1.34)	0.714	0.512 (0.46;0.56)	0.001	0.502 (0.41;0.61)	0.001
**Female patient (vs. male)**	—	—	**1.316 (1.23;1.41)**	**0.001**	**1.195 (1.15;1.24)**	**0.001**	—	—
**Patient’s age (cont.)**	—	—	**0.979 (0.98;0.98)**	**0.001**	**0.991 (0.99;0.99)**	**0.001**	**0.985 (0.98;0.99)**	**0.001**
**Patient’s place of origin**	—	—	—	—	—	—	—	—
**Spain**	Ref.		Ref.		Ref.		Ref.	
**Americas**	1.264 (0.92;1.75)	0.155	**1.257 (1.06;1.49)**	**0.008**	**1.179 (1.08;1.29)**	**0.001**	**1.441 (1.30;1.60)**	**0.001**
**Asia/Oceania**	1.570 (0.97;2.54)	0.066	**1.562 (1.16;2.11)**	**0.004**	**1.398 (1.20;1.63)**	**0.001**	**1.258 (1.02;1.55)**	**0.032**
**Europe outside Spain**	1.388 (0.86;2.24)	0.179	**1.321 (1.05;1.66)**	**0.016**	**1.434 (1.28;1.61)**	**0.001**	**1.522 (1.32;1.75)**	**0.001**
**Africa**	**1.756 (1.30;2.38)**	**0.001**	**1.312 (1.05;1.63)**	**0.015**	1.095 (0.98;1.22)	0.108	**1.218 (1.05;1.42)**	**0.010**
**BMI (cont.)**	**0.980 (0.97;0.99)**	**0.001**	0.997 (0.99;1.00)	0.286	—	—	**0.993 (0.99;1.00)**	**0.011**
**≥1 n** **ew general prescriptions^a^** **(vs. none)**	**1.223 (1.05;1.43)**	**0.011**	**1.229 (1.14;1.33)**	**0.001**	0.895 (0.78;1.03)	0.114	—	—
**Number of visits ^a^ (cont.)**								
**Visits to GP**	—	—	—	—	—	—	**0.992 (0.99;1.00)**	**0.007**
**Visits to nurse**	—	—	—	—	—	—	0.997 (0.99;1.00)	0.218
**Active illnesses**								
**Pain**	0.897 (0.79;1.02)	0.096	—	—	—	—	**1.056 (1.00;1.11)**	**0.040**
**Respiratory**	—	—	0.990 (0.90;1.09)	0.839	—	—	0.970 (0.89;1.06)	0.507
**Physical disability ^b^**	—	—	1.002 (0.93;1.08)	0.955	1.042 (1.00;1.09)	0.077	0.956 (0.90;1.02)	0.175
**Cardiovascular conditions**								
**Hypertension**	0.879 (0.77;1.01)	0.062	**0.773 (0.72;0.83)**	**0.001**	**0.751 (0.72;0.78)**	**0.001**	**0.784 (0.74;0.83)**	**0.001**
**Dyslipidemia**	**0.825 (0.72;0.94)**	**0.004**	**0.869 (0.81;0.93)**	**0.001**	**0.895 (0.86;0.94)**	**0.001**	—	—
**Recent CVD (≤6 months)**	—	—	**0.558 (0.49;0.64)**	**0.001**	**0.861 (0.76;0.98)**	**0.022**	**0.709 (0.59;0.84)**	**0.001**
**Established CVD (>6 months)**	—	—	**0.680 (0.61;0.75)**	**0.001**	1.058 (0.99;1.13)	0.081	1.050 (0.96;1.15)	0.310
**Mental disorders**	—	—	**0.847 (0.79;0.91)**	**0.001**	—	—	—	—
**Diabetes (1 and 2)**	—	—	**0.658 (0.61;0.71)**	**0.001**	**0.735 (0.70;0.77)**	**0.001**	**0.826 (0.77;0.89)**	**0.001**
**Digestive system disorder**	—	—	1.025 (0.95;1.11)	0.550	—	—	0.973 (0.91;1.04)	0.423
**Urticaria/allergy**	—	—	—	—	**1.157 (1.03;1.30)**	**0.017**	—	—
**Female GP (vs. male)**	—	—	—	—	—	—	**0.876 (0.83;0.92)**	**0.001**
**Substitute/resident GP (vs. other)**	—	—	**1.282 (1.11;1.48)**	**0.001**	**1.320 (1.21;1.44)**	**0.001**	**1.312 (1.18;1.46)**	**0.001**
**Teaching PC center (vs. regular)**	—	—	**0.901 (0.83;0.98)**	**0.015**	—	—	**0.888 (0.83;0.95)**	**0.001**

ACEI: Angiotensin-converting enzyme inhibitors; cont.: continuous variable; CVD: Cardiovascular disease; GP: General practitioner; SES: Socioeconomic status. **Bold** numbers indicate statistically significant results. “—” indicates that the factor was not selected to be included in the multivariate analysis based on results from the bivariate analysis. ^a^ In the 12 months preceding the index prescription. ^b^ Physical disabilities include blindness, urinary incontinency and hypoacusia.

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
