# Peer review of "Initiation and Single Dispensing in Cardiovascular and Insulin Medications: Prevalence and Explanatory Factors"

_ijerph, 2020, doi:10.3390/ijerph17103358_

Round 1

Reviewer 1 Report

In general, writing must be improved. The information is not well presented, tables are confusing. For example, SD is shown in parenthesis (), and that presentation is confusing.

The conclusions are adequate, but the information is not well presented, for that it is difficult to follow the author's narrative.

  1. it is necessary to support where this conclusion comes from the study (abstract)
  2. Where the authors wrote " Known Factors" in the abstract, which factors?? the conclusion must be specific.
  3. In the introduction section, there are a lot of references, most of them are no relevant for the study, and they are old.
  4. The method section did not present technical details, for example it is missing the used Softwares where the analysis was made
  5. In Table 1 put a ±  when you are considering SD because it is confusing to observe the data in the present format.
  6. Table 3 and 4 is so big and confusing, the information should be divided or make some graphs where the variables and relations could be observed. 
  7. The discussion section is the best part, but it is not easy to follow the authors because previous sections are confusing 
  8. As I mentioned before, there are a lot of references, some of them are no relevant for the study, especially some of the introduction section, and they are old. The article needs to be rewritten with newer references and focused on relevant information.

I made specific notes that you will find attached. 

Author Response

Reviewer: In general, writing must be improved. The information is not well presented, tables are confusing. For example, SD is shown in parenthesis (), and that presentation is confusing.

Answer: We reviewed the paper and sent it to an English editing service to improve the writing. We also modified the tables and included a ± symbol before SD to avoid confusion.

The conclusions are adequate, but the information is not well presented, for that it is difficult to follow the author's narrative. it is necessary to support where this conclusion comes from the study (abstract) Where the authors wrote " Known Factors" in the abstract, which factors?? the conclusion must be specific.

A: We shortened the methods section in the abstract and made presented conclusions in a more detailed way.

In the introduction section, there are a lot of references, most of them are no relevant for the study, and they are old.

A: We have substantially reduced the number of references, especially those less updated. The numbers of references were reduced from 79 to 49.

The method section did not present technical details, for example it is missing the used Softwares where the analysis was made

A: Thank your for noticing this. It is now detailed in line 246 at the end of method section.

“All analyses were performed using STATA MP 13.0

In Table 1 put a ±  when you are considering SD because it is confusing to observe the data in the present format.

A: Thank you for this suggestion. Tables look much more clearer now.

Table 3 and 4 is so big and confusing, the information should be divided or make some graphs where the variables and relations could be observed.

A: We agree with reviewer and we are aware of that. Figure 1 summarizes all the information presented in Table 1. However, we consider crucial (for transparency) to show the specific estimates. We are convinced that final print version will let observe these tables in a better way than its current form.

The discussion section is the best part, but it is not easy to follow the authors because previous sections are confusing

As I mentioned before, there are a lot of references, some of them are no relevant for the study, especially some of the introduction section, and they are old. The article needs to be rewritten with newer references and focused on relevant information.

A: We expect that the changes introduced contributed to make the discussion more easy to follow.

I made specific notes that you will find attached.

A: There was only one note that was referred to how is shown SD in tables. It was already answer in a previous comment.

Reviewer 2 Report

The paper is very interesting and present data that are not usually reported regarding the lack of adherence at initiation. 

  • In section 2.2 Design (and in the abstract) the authors define this study as a retrospective observational study. According to STROBE guidelines, The STROBE guidelines do not allow the use of the words “prospective” or “retrospective” or “concurrent” or “historical,” but rather encourage the researcher to describe the actual methodology. We recommend that, whenever authors use these words, they define what they mean. Most importantly, we recommend that authors describe exactly how and when data collection took place (Vandenbroucke et al. 10.1371/journal.pmed.0040297).  Most RWD are retrospective when we analyze them, but they have been collected prospectively, so I strongly recommend not to use this term. 
  • Section 2.2, line 144. Correct spelling REGICOR instead of RECICOR
  • Table 1.
    • It's only a suggestion, but for some readers, an overall column is sometimes useful. 
    • Are there missing data for MEDEA or BMI? They should be reported. 
  • Table 3, 4. Have you considered using BMI with categories instead of a continuous variable? And have you captured the obesity as an ICD-10 diagnosis code?
  • Discussion, lines 359-360. For some patients after a CV event, only antiplatelets, beta-blocker and statins are recommended, while ACEI is not recommended for all patients. So is it possible that the lack of ACEI adherence is caused by this recommendations? (See Ibañez B, doi:10.1093/eurheartj/ehx393, and Barrabés J, doi:10.1016/j.recesp.2015.11.001). 

Author Response

The paper is very interesting and present data that are not usually reported regarding the lack of adherence at initiation.

A: Thank you very much for your kind comment.

In section 2.2 Design (and in the abstract) the authors define this study as a retrospective observational study. According to STROBE guidelines, The STROBE guidelines do not allow the use of the words “prospective” or “retrospective” or “concurrent” or “historical,” but rather encourage the researcher to describe the actual methodology. We recommend that, whenever authors use these words, they define what they mean. Most importantly, we recommend that authors describe exactly how and when data collection took place (Vandenbroucke et al. 10.1371/journal.pmed.0040297).  Most RWD are retrospective when we analyze them, but they have been collected prospectively, so I strongly recommend not to use this term.

A: We agree with the reviewer. As  STROBE guidelines suggest, the word retrospective may imply confusion, consequently we have deleted  the word “retrospective” (L137). We gave further details on the process of data collection in lines 144 and 145.

“The SIDIAP database is managed by public healthcare authorities, is anonymous, encoded and secure, and meets all current legal requirements. All the information used for analysis purposes was recorded between 2011 and 2014 and it was gathered from the database in 2015. This information contains patient, GP, and PC center information, along with information on prescription and whether or not a given prescription has been dispensed.”

Section 2.2, line 144. Correct spelling REGICOR instead of RECICOR

A: It is now corrected, thank you.

Table 1.

It's only a suggestion, but for some readers, an overall column is sometimes useful.

A: Thank you for the suggestion. We also considered this but decided not to include it because all the analyses and results are presented separately for the 4 medication groups and Tables are large.

Are there missing data for MEDEA or BMI? They should be reported.

A: Overall missing data is detailed in 2.4 Analysis section (L220-223).

To manage missing data (BMI (30.8%), prescriber sex (9.7%), socioeconomic status (4.2%), and place of origin (41%)), simple imputation by chained equations was used with logistic regression and ordinal logistic models using all the available data (2011-2014).(25) This imputation method has been previously used with satisfactory results.(21,25–27) For BMI, a truncated regression model with a lower limit of 10 was used.

Table 3, 4. Have you considered using BMI with categories instead of a continuous variable? And have you captured the obesity as an ICD-10 diagnosis code?

A: We considered to use BMI as a continuous variable because we thought it gives us more information. Unfortunately, we did not capture obesity as ICD-10 diagnosis code.

Discussion, lines 359-360. For some patients after a CV event, only antiplatelets, beta-blocker and statins are recommended, while ACEI is not recommended for all patients. So is it possible that the lack of ACEI adherence is caused by this recommendations? (See Ibañez B, doi:10.1093/eurheartj/ehx393, and Barrabés J, doi:10.1016/j.recesp.2015.11.001).

A: This is a very interesting suggestion. We included it to the paper (lines 409-418) and will take it into account when designing interventions.

Another contradictory result is that, whereas having a recent cardiovascular disease increased the risk of non-initiation of ACEIs, it decreased the risk of non-initiation of statins. Previous studies have reported lower levels of adherence to ACEIs than to statins after a cardiovascular event.(38) The patient may underestimate the danger of hypertension and fail to link it to cardiovascular disease.(39) Another possible explanation is that the patient is aware that ACEIs are generally not recommended after a cardiovascular event.(40). To confirm this finding, future studies should explore the influence of a recently established cardiovascular disease, and qualitative studies with patients should be conducted in order to better understand the influence of cardiovascular disease on initiation.

Reviewer 3 Report

The manuscript “Initiation and single dispensing in cardiovascular and insulin medications: prevalence and explanatory factors” sought to assess the explanatory factors and prevalence of non-initiation and single dispensation of cardiovascular medication and insulin prescribed in primary care in Catalonia. This manuscript was very well written and has a novelty in discovering BMI and concomitant diseases as new explanatory factors for non-initiation and single dispensing. Also, a though discussion further helped the readers to better understand the problem and to raise awareness of the importance of following through prescription. As a result, very minor editorial changes are needed.

  1. Page 7, Line 260, Figure 1 was already mentioned. Then the actual figure was only shown on page 12. The authors can consider put figure 1 in section 3.2.

  1. Page 13, Line 357-360, “Another contradictory result is that, whereas having a recent cardiovascular disease increased the risk of non-initiation of ACEIs, it decreased the risk of non-initiation of statins. Previous studies have reported lower levels of adherence to ACEIs than to statins after a cardiovascular event”. Is there a consideration that patients might think statin as a safer option compared to ACEIs, as well as the commercialization (ads on TV or by word of mouth) of each drug in the area.

Author Response

The manuscript “Initiation and single dispensing in cardiovascular and insulin medications: prevalence and explanatory factors” sought to assess the explanatory factors and prevalence of non-initiation and single dispensation of cardiovascular medication and insulin prescribed in primary care in Catalonia. This manuscript was very well written and has a novelty in discovering BMI and concomitant diseases as new explanatory factors for non-initiation and single dispensing. Also, a though discussion further helped the readers to better understand the problem and to raise awareness of the importance of following through prescription. As a result, very minor editorial changes are needed.

Answer: Thank you for this kind comment.

Page 7, Line 260, Figure 1 was already mentioned. Then the actual figure was only shown on page 12. The authors can consider put figure 1 in section 3.2.

A: We agree with the reviewer, however this was an editorial decision. We expect that the editorial office takes this comment into consideration. Thank you for this.

Page 13, Line 357-360, “Another contradictory result is that, whereas having a recent cardiovascular disease increased the risk of non-initiation of ACEIs, it decreased the risk of non-initiation of statins. Previous studies have reported lower levels of adherence to ACEIs than to statins after a cardiovascular event”. Is there a consideration that patients might think statin as a safer option compared to ACEIs, as well as the commercialization (ads on TV or by word of mouth) of each drug in the area.

A: The Spanish regulations do not allow commercials/advertisements on medicines. Therefore, this could not explain the result in our context.